



# A long-term (2000-2020) global 0.05 ° continuous atmospheric carbon dioxide dataset (GCXCO₂) combining OCO-2 observations and model simulations based on stack learning

Xiaobin Guan[1], Zhihao Sun[1], Dong Chu[2], Guanglei Xie[1], Yuchen Wang[1], Huanfeng Shen[1,3]

[1]School of Resource and Environmental Sciences, Wuhan University, Wuhan 430079, China
[2]Key Laboratory of Earth Surface Processes and Regional Response in the Yangtze-Huaihe River Basin, Anhui Province, School of Geography and Tourism, Anhui Normal University, 241002, China
[3]Collaborative Innovation Center of Geospatial Technology, Wuhan, China

*Correspondence to*: Huanfeng Shen (shenhf@whu.edu.cn)

**Abstract.** High-accuracy atmospheric (carbon dioxide) $CO_2$ concentration data are critical in understanding the global carbon cycle, but there is still a lack of a high-resolution $CO_2$ product with long-term and global seamless coverage. In this study, a global continuous 8-day $XCO_2$ (column-averaged $CO_2$ dry air mole fraction) product (GCXCO₂) was reconstructed at a spatial resolution of 0.05 ° from 2000 to 2020, based on OCO-2 satellite data. An ensemble machine learning stacking regression model, which combines light gradient boosting machine (LGBM), extreme gradient boosting (XGB), extremely randomized trees (ETR), gradient boosting regression (GBR), and random forest (RF), was utilized to model the relationships between $XCO_2$ data and auxiliary satellite, simulation data, and meteorological data. A dynamic normalization strategy was developed to handle the great temporal variation issue and ensure the temporal expansion of the prediction model. Multiple validation methods were applied to comprehensively evaluate the spatial and temporal generalization ability of the model and product. The 10-fold cross-validation shows an overall satisfactory result at a global scale, with $R^2 = 0.974$ and root-mean-square error (RMSE) = 0.551 ppm (parts per million). Further spatial extension and temporal prediction experiments also proved that dependable results could be obtained in the regions and time periods without valid OCO-2 satellite observations ($R^2 = 0.958$ and $R^2 = 0.886$, respectively). Compared with Total Carbon Column Observing Network (TCCON) ground station observations, the GCXCO₂ product performs better than the model simulation data, demonstrating a better accuracy and a higher spatial resolution. Based on the GCXCO₂ product, an upward annual trend of approximately 2.09 ppm/year can be found for global $XCO_2$ between 2000 and 2020, and significant differences are found between the Northern and Southern hemispheres in different seasons. This product may well be the first remote sensing-based global high-precision long-term $XCO_2$ dataset, which will help advance the understanding of climate change and carbon balance. The dataset can be obtained freely at https://doi.org/10.5281/zenodo.10083102 (Guan and Sun, 2023).



## 1 Introduction

The continuous increase of greenhouse gases in the atmosphere has already induced severe global climate change problems (Hegerl and Cubasch, 1996; Wuebbles and Jain, 2001; Lioubimtseva and Adams, 2004; Lonngren and Bai, 2008; Zhang and Caldeira, 2015) and significantly impacted human well-being (Tagwi, 2022). Carbon dioxide ($CO_2$) is one of the main greenhouse gases, and the global average $CO_2$ has increased from 336.85 ppm in 1979 to 417.06 ppm in 2022, according to the National Oceanic and Atmospheric Administration (NOAA). Therefore, high-precision quantitative assessment of global $CO_2$ concentration is crucial for addressing the constantly changing situation.

At present, $CO_2$ column concentration data are obtained based on three main methods: ground station observations, model simulation, and satellite estimation. Ground stations usually use a Fourier transform spectrometer (FTS) to directly measure solar radiation in the near-infrared band, thereby inverting the concentration of $CO_2$ without the effects of aerosols and clouds. This method, as used by the Total Carbon Column Observing Network (TCCON), can observe column-averaged $CO_2$ dry air mole fraction ($XCO_2$) with a high accuracy and low uncertainty, but is usually limited by the sparse distribution of stations and the fact that it is difficult to conduct $CO_2$ monitoring in large regions. Model simulation methods consider the physical, chemical, and biological processes of $CO_2$, and estimates its concentration and carbon flux through an atmospheric transport model (Krol et al., 2005), such as CarbonTracker (CT), the Copernicus Atmosphere Monitoring Service (CAMS), and the Global Carbon Assimilation System (GCASv2) (Jiang et al., 2021). By assimilating carbon emission inventories and $CO_2$ observation data, the model simulation $XCO_2$ usually shows a relatively high accuracy at the intercontinental scale (Kong et al., 2019), but its spatial resolution is too coarse for regional applications (Mustafa et al., 2020). For example, the spatial resolution of the CT dataset is only $3 \times 2$ degrees, and the spatial resolution of the CAMS global greenhouse gas reanalysis (EGG4) dataset is $0.75 \times 0.75$ degrees.

In recent years, satellite remote sensing based estimation has become a new way to obtain $XCO_2$ data with a higher spatial resolution, and a series of satellite products have been published based on various sensors. The satellites used for monitoring the global distribution of $CO_2$ include the Greenhouse Gases Observing Satellite (GOSAT) (Yokota et al., 2009) and the GOSAT-2 satellite, which were launched in 2009 and 2018 by Japan, respectively. The United States launched the Orbiting Carbon Observatory-2 (OCO-2) satellite in 2014 (Eldering et al., 2017) and the OCO-3 satellite in 2019 (Eldering et al., 2019). China launched the TanSat satellite in 2016 (Ran and Li, 2019). Satellite observation from space-based platforms can achieve high-resolution repeated observations, and thus timely and accurate detection of changes in $XCO_2$ can be achieved (Liu et al., 2020). However, due to the satellite orbit and observation angle limitations, there are serious missing data problems in the current satellite products. As shown in Fig. 1, all the observations of the OCO-2 satellite over one month show a strip-shaped pattern with apparent gaps, and there are only a few observations in high-latitude areas. These issues make it almost impossible to monitor global $CO_2$ concentration and carbon flux using only remote sensing data.

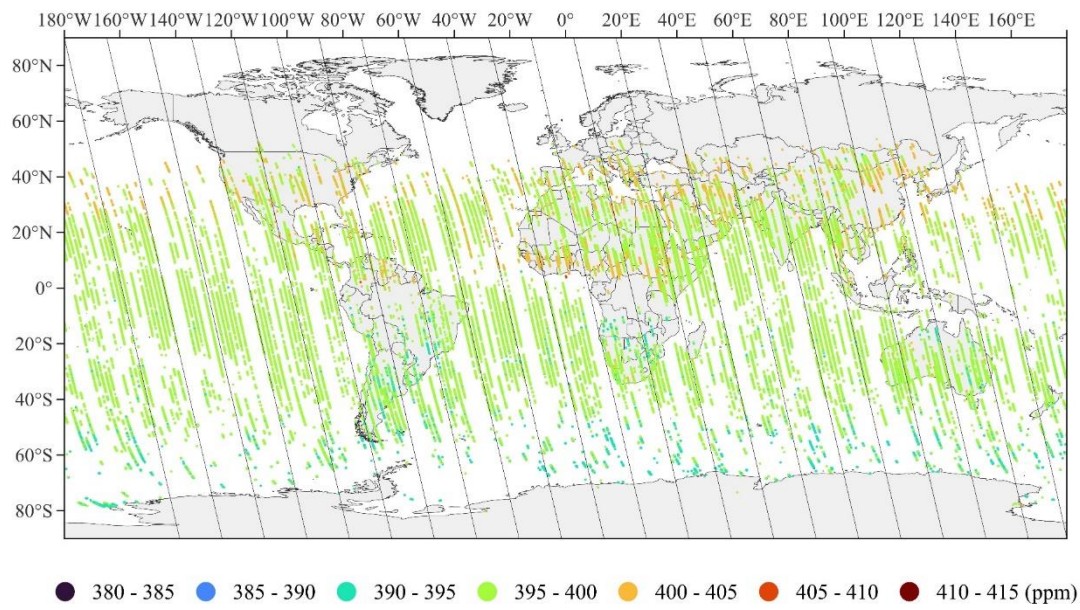

**Figure 1.** The distribution of the OCO-2 satellite observations for January 2015.

As a result, a series of seamless mapping methods have been developed in recent years, in order to solve the missing data issue of the satellite observations and obtain continuous $XCO_2$ data. These methods can be divided into three main types: reconstruction-based methods, fusion-based methods, and data-driven methods. The reconstruction-based methods mainly consider the spatial and temporal continuity and correlation of the $XCO_2$ distribution in the product itself to fill the gaps, and thus they do not require any other auxiliary data (Yue et al., 2015; He et al., 2020; Ma et al., 2021). For this reason, these methods are easy to implement, but they cannot reconstruct areas well that have sparse satellite observations. The fusion-based methods integrate multiple data sources, including satellite data (Jing et al., 2014; Jin et al., 2022) and model simulation data (Mingwei et al., 2017; Sheng et al., 2022; Liang et al., 2023), to obtain seamless $XCO_2$. These methods can integrate multi-source observations to obtain seamless data with a stable accuracy, but the spatial resolution is still limited. Over the last two years, data-driven methods have become a popular way to obtain continuous $XCO_2$ data by establishing the relationships between $XCO_2$ and related explanatory variables (Li et al., 2022; Zhang and Liu, 2023). Machine learning is the most widely used method, which has a strong nonlinear fitting capability, and can thus achieve a higher precision than the other methods (He et al., 2022; Li et al., 2022; Zhang et al., 2022; Zhang and Liu, 2023). Based on machine learning, several $XCO_2$ datasets have been produced and the spatio-temporal variation has been analyzed in different regions.

Although previous studies have already produced several global products, there are still obvious limitations. First of all, most of the current global coverage products only focus on the $XCO_2$ mapping in terrestrial areas, and the ocean areas are neglected. As a result, this is still not globally continuous mapping and cannot meet the demands of global carbon change research. This may be due to the abundance of explanatory variables in terrestrial areas, while there is a lack of such variables in ocean



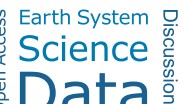

regions. Secondly, a true long-term global $XCO_2$ product is still lacking, and most of the previous studies have only
reconstructed the years in which satellite data are available. Therefore, only a few years of data can be used for long-term
analysis. This may be due to the significant changes in $XCO_2$, making it difficult for the model to expand in the temporal
dimension. Finally, the previously produced $XCO_2$ products have still not been well validated, with the spatial and temporal
extension capacity overlooked. Although the commonly used 10-fold cross-validation and ground station data can evaluate the
quantitative performance of the model, the accuracy in the regions and times without satellite observations is not well assessed.
In summary, the current $XCO_2$ products cannot truly achieve long-term global coverage and have not verified the accuracy of
areas without satellite observations. It is therefore necessary to develop new $XCO_2$ mapping methods to overcome these
shortcomings.
Therefore, the aim of this study was to produce a novel global seamless $XCO_2$ product (GCXCO$_2$) with a long temporal
coverage and high spatio-temporal resolution, based on a machine learning method. The main objectives of this study were: 1)
to develop a true global seamless $XCO_2$ mapping method based on ensemble machine learning, covering both terrestrial and
ocean areas; 2) to comprehensively evaluate the spatio-temporal stability of the model and product based on various validation
methods; and 3) to analyze the global $XCO_2$ distribution and variation characteristics in different seasons and years, based on
this product.

## 2 Material and methods

### 2.1 Data sources

#### 2.1.1 OCO-2 satellite data

The OCO-2 satellite uses three-channel high-resolution imaging grating spectrometers to measure the reflected sunlight in the
short-wave-infrared (SWIR) $CO_2$ bands and in the near-infrared (NIR) molecular oxygen ($O_2$) A band (Oyafuso et al., 2017),
with a revisit period of 16 days and an equator crossing time of approximately 1:30 pm. The OCO-2 satellite cross-slit width
is approximately 1.29 km at nadir, with 2.25 km in footprint length along-track. The OCO-2 version 10 Level 2 Full Physics
(OCO2_L2_Lite_FP_10r) products from 2015 to 2020 were used in this study, which can provide daily $XCO_2$, solar-induced
fluorescence (SIF), and other atmospheric surface properties after radiometric correction. The $XCO_2$ variable with high quality
(flag = 0) in the products was selected and aggregated into regular grid data with a spatial resolution of 0.05 degrees and a
temporal resolution of 8 days. We selected the grid cells with more than 10 observations during a period and took the mean
value as the observation value.

#### 2.1.2 TCCON data

The TCCON, which was established in 2004, is a global greenhouse gas observation network based on FTSs (Toon et al.,
2009), mainly monitoring gases such as $CO_2$, methane ($CH_4$), and nitrous oxide ($N_2O$) in the atmosphere (Yang et al., 2020).





The direct solar spectra are measured in the NIR band to retrieve the column abundances of these gases. Currently, the TCCON
has a total of 30 operating stations around the world, with five stations no longer operating and four potential future stations.
In this study, version GGG2020 data (https://tccondata.org/) were used, and the observation data with a fractional variation in
solar intensity (FVSI) value of more than 5% were filtered out. There are 30 stations with observation records covering 2004
to 2020. The location of each station used in this study is shown in Fig. 2. It is clear that all the stations are located in land
areas, mainly distributed in North America, Europe, and East Asia in the Northern Hemisphere, and rarely in the Southern
Hemisphere.

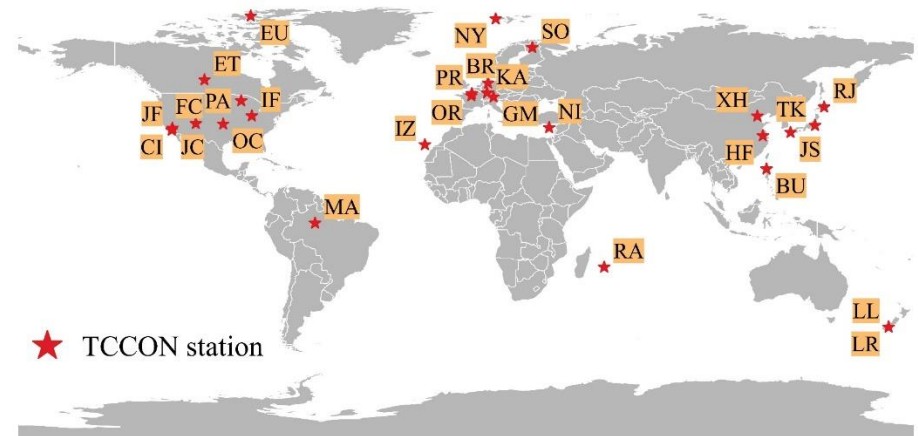


**Figure 2.** The distribution of the TCCON stations used in this study.
**2.1.3 Remote sensing auxiliary data**
The remote sensing auxiliary data used in this study were the Enhanced Vegetation Index (EVI), chlorophyll-a (CHL-a) data,
and Moderate-resolution Imaging Spectroradiometer (MODIS) land surface reflectance data. Vegetation plays a critical role
in $CO_2$ absorption in ecosystems (Vicca, 2018), but there is a lack of variables that measure both terrestrial and ocean vegetation.
Therefore, different key variables were selected in this study, with EVI and CHL-a representing the $CO_2$ uptake capacity of
land and ocean, respectively. MODIS reflectance band 6 (1628–1652 nm) and band 7 (2105–2155 nm) data, which are close
to the observation band of the OCO-2 satellite, were also utilized. The reflectance and EVI data can be downloaded from
(https://ladsweb.modaps.eosdis.nasa.gov/), and the CHL-a data can be obtained from the Ocean Biology Processing Group
(https://oceancolor.gsfc.nasa.gov/). In order to process the missing EVI, CHL-a, and reflectance data, the adaptive spatio-
temporal tensor completion (ST-Tensor) method (Chu et al., 2021) was applied to reconstruct the global seamless remote
sensing auxiliary data. The EVI and CHL-a variables were both normalized to generate the fused CHLEVI variable, which
can effectively represent the global $CO_2$ absorption capacity.



### 2.1.4 Model simulation data

NOAA's CarbonTracker version CT2022 dataset (Jacobson et al., 2023) and the CAMS EGG4 dataset were used as $XCO_2$ model simulation data and to provide the initial spatial distribution of $CO_2$. These datasets can be downloaded from the Global Monitoring Laboratory (https://gml.noaa.gov/) and (https://ads.atmosphere.copernicus.eu/), respectively. CarbonTracker is a $CO_2$ measurement and modeling system, which is used to track the global carbon sources and sinks. It should be noted that the CT2022 dataset assimilates data from 559 stations provided by 66 laboratories around the world, and the data have been adjusted for the changes in fossil fuel emissions caused by the COVID-19 pandemic. In contrast, the EGG4 dataset focuses on the greenhouse gases of $CO_2$ and $CH_4$ and assimilates the observation data of the GOSAT, Envisat, MetOp-A, and MetOp-B satellites. However, the data have not been adjusted for the effect of the COVID-19 pandemic. This may have led to significant deviations in the $XCO_2$ simulated by CAMS for several years around 2019. Therefore, the utilization of two $XCO_2$ model simulation datasets can provide more information and reduce the dependence of the results on a single set of model simulation data.

### 2.1.5 Meteorological data

The meteorological data selected for the modeling were obtained from the Modern-Era Retrospective analysis for Research and Applications, Version 2 (MERRA-2). MERRA-2 is the first long-term global reanalysis to assimilate space-based observations of aerosols, and provides data from 1980 to the present (Gelaro et al., 2017). In this study, global continuous meteorological variables were utilized to establish nonlinear relationships with $XCO_2$, including air temperature (TEM), wind (U component, V component), specific humidity (QV), sea level pressure (SLP), and surface incoming shortwave flux (SWGDN). These variables are fundamental factors affecting atmospheric transport and vegetation growth, which can be downloaded from (https://disc.gsfc.nasa.gov/datasets?keywords=merra2&page=1).

The remote sensing auxiliary data, model simulation data, and meteorological data were all auxiliary data for the model input, and were resampled with spatial and temporal resolutions of 0.05 degrees and 8 days, respectively. The detailed information of the variables can be found in Table 1.





**Table 1.** The detailed information of the auxiliary data used in this study.

| Data type | Source | Variable | Temporal resolution | Spatial resolution |
|---|---|---|---|---|
| Remote sensing data | MOD09CMG | Reflectance | Daily | 0.05° |
| | MOD13C1 | EVI | 16 days | 0.05° |
| | MODIS OBPG | CHL-a | 8 days | 0.05° |
| $CO_2$ simulation data | CT2022 | CT | 3 h | 3°× 2° |
| | CAMS EGG4 | CAMS | 3 h | 0.75°×0.75° |
| Meteorological data | MERRA-2 | TEM | Daily | 0.5°×0.625° |
| | | QV | | |
| | | Wind (U,V) | | |
| | | SLP | | |
| | | SWGDN | | |

**2.2 Model description**
**2.2.1 Overall workflow**
The long-term global continuous $XCO_2$ mapping process can be divided into three steps (Fig. 3): data processing, model
training and validation, and $XCO_2$ mapping and spatio-temporal analysis.
**Step 1:** Data processing. By using seamless auxiliary data to match the gridded OCO-2 satellite data, a total of 4833846 data
were obtained for 2015 to 2020. The OCO-2 satellite observations were regarded as the true values, and the other variables
were used as explanatory variables. Due to the continuous increase of $XCO_2$ from 2000 to 2020, if we directly trained the
model with the OCO-2 satellite $XCO_2$ as the true values, there would be an out-of-range problem during the model prediction,
which means that the model had not learned the corresponding $CO_2$ concentration.
A dynamic normalization strategy was introduced to address this issue. The $XCO_2$ normalization was implemented separately
for each period so that the model labels were not limited by the $XCO_2$ range, which is the reason why this is called dynamic
normalization. Specifically, we calculated the maximum and minimum values of the global model simulation CT values for
each period, with the maximum value multiplied by 1.02 (as MAX) and the minimum value multiplied by 0.98 (as MIN). Then
the CT, CAMS, and OCO-2 satellite observation $XCO_2$ data were normalized to 0–1 using MAX and MIN.
**Step 2:** Model training and validation. Based on dynamic normalization, 150000 OCO-2 matched data from 2015 to 2018
were selected at random to establish the nonlinear relationships between the $XCO_2$ and auxiliary data. The ensemble machine
learning stacking regression model was selected for the modeling. For the trained stacking regression model, various validation
methods were designed to validate the spatial and temporal generalization ability, including 10-fold cross-validation, spatial
expansibility validation, temporal extension validation, and ground station data validation.


**Step 3:** Mapping and spatio-temporal analysis. We produced the global XCO$_2$ product with full coverage of 0.05 degrees every
8 days from 2000 to 2020. The model trained in Step 2 was used to produce the global maps from 2003 to 2020. Due to the
lack of CAMS and CHLEVI from 2000 to 2002, we removed these variables and trained another model for the mapping from
2000 to 2002. The corresponding model validation results are included in the supplementary material (Figs. S1–S5). Based on
the product for 2000 to 2020, the spatial distribution characteristics of XCO$_2$ in different seasons and years were explored. At
the same time, the change trends of XCO$_2$ at the global scale and different latitude areas were analyzed.

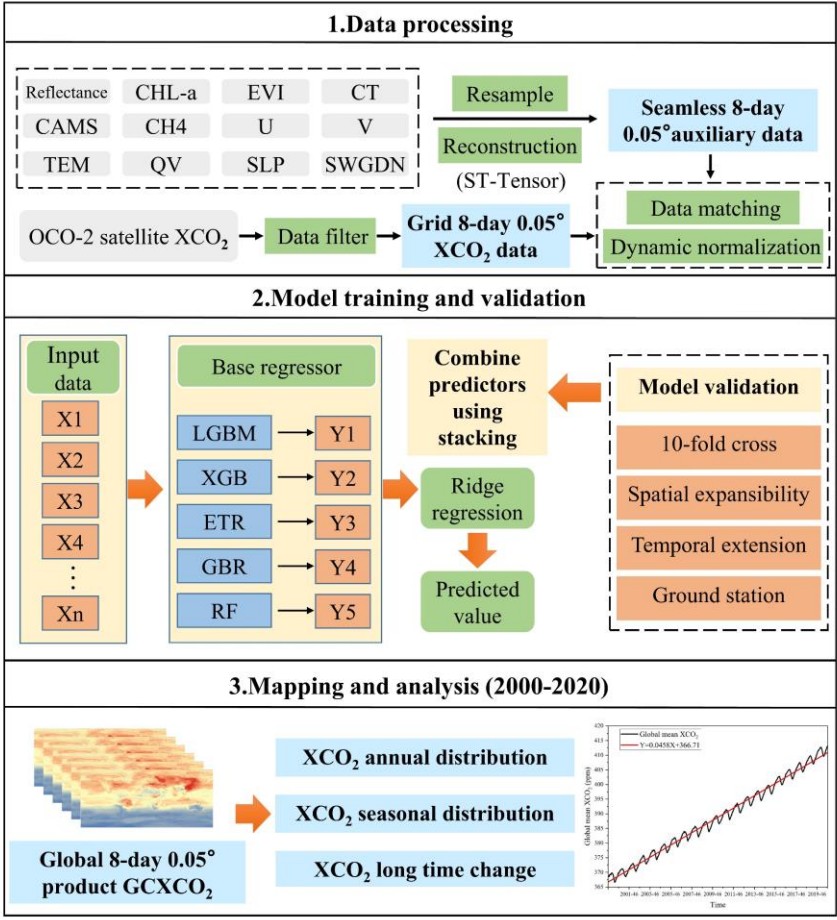


**Figure 3.** The overall workflow of this study.
**2.2.2 Stacking regression model**
Stacking regression (Wolpert, 1992) is an ensemble machine learning method that combines multiple basic regression models
with a meta-regression model, which can minimize the error rate of the multiple regression models. The model structure
typically consists of two layers, with the first layer containing the many basic regression models and the second layer
containing the meta-regression model. For the input variables, each basic model predicts a value and inputs the predicted value

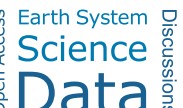

into the meta model to compute the final prediction. Previous studies (Sesmero et al., 2015) have shown that the stacking is an
effective way to improve the performance of the model. In general, it is necessary to choose regressors with significant
differences in the first layer, in order to combine different model characteristics. Meanwhile, simple regression model is usually
selected for the second layer (Ting and Witten, 1997), to prevent overfitting of the model.
In this study, five basic models were selected in the first layer of the stacking regression model (Fig. 3): light gradient boosting
machine (LGBM) regression, extreme gradient boosting (XGB) regression, extremely randomized trees (ETR) regression,
gradient boosting regression (GBR), and random forest (RF) regression. These models all perform well and have different
characteristics. LGBM, XGB, and GBR are boosting models that continuously improve on the weak regressors, but their
improvement strategies are different. ETR and RF are bagging models that use multiple independent decision trees for the
regression, but the splitting methods for the tree nodes are different. In the second layer, we selected ridge regression model
to deal with the multicollinearity problem of the first layer output value. Compared with ordinary linear regression, the ridge
regression (Hoerl and Kennard, 1970) adds L2 regularization constraint to the coefficient of loss function, which can avoid the
significant change of coefficient and make the regression model more stable. Based on the stacking regression model, the
nonlinear relationships between $XCO_2$ and the explanatory variables were constructed as shown in Eq. (1) :
$$OCO2\_N = f(TEM, QV, U, V, SLP, SWGDN, REF6, REF7, CT\_N, CAMS\_N, CHLEVI, SCYCLE) \tag{1}$$
In the equation, the meaning of auxiliary variables such as TEM can be found in Section 2.1.3 to Section 2.1.5. REF6 and
REF7 represent the MODIS reflectance band 6 and band 7 data, respectively. OCO2_N, CT_N, and CAMS_N represent the
corresponding normalized variables, and f refers to the nonlinear relationships built on the stacking regression model.
Considering the periodicity and seasonality of $XCO_2$ variation (Zhang and Liu, 2023), the SCYCLE variable was designed to
describe this characteristic, which is equal to the sine value of the cycle of one year, which is calculated as shown in Eq. (2)
(the range of cycle values is 1 to 46) :
$$SCYCLE = \sin(cycle * \pi/23) \tag{2}$$

**2.3 Model validation**

Typically, 10-fold cross-validation and ground station data have been widely used in past studies to evaluate the model.
However, these methods cannot fully validate the spatial and temporal generalization ability and the decay performance. In
this study, two more validation methods were designed to sufficiently evaluate the model's spatio-temporal performance, i.e.,
spatial expansibility validation and temporal extension validation. For all of the validation methods, we calculated the $R^2$ and
root-mean-square error (RMSE) as the evaluation indicators. It should be noted that we denormalized the output of the stacking
regression model by using the MAX and MIN of the corresponding period when calculating the evaluation indicators. The
specific meanings of the four validation methods are as follows.



### 2.3.1 Ten-fold cross-validation

In general, 10-fold cross-validation (Breiman and Spector, 1992) can evaluate the model's accuracy on the whole dataset and
determine whether the model is overfitting. In this study, the matched OCO-2 data from 2015 to 2018 were randomly divided
into 10 subsets to validate the stacking regression model. The 10-fold cross-validation uses nine subsets to train the model and
one subset to test the model each time, and repeats this operation 10 times to test each subset.

### 2.3.2 Spatial expansibility validation

The distribution of OCO-2 observations is very sparse, with many areas without samples, and the accuracy of these areas has
not been validated by 10-fold cross-validation. Therefore, spatial expansibility validation was designed to evaluate the spatial
generalization ability of the stacking regression model. The global area was divided into 23 regions according to the shape of
the OCO-2 satellite observation bands (Fig. 1, the solid lines). Similar to the 10-fold cross-validation, the matched OCO-2 data
between 2015 and 2018 from each region were used separately for the validation, while 150000 data were randomly selected
from other regions to train the stacking regression model. Based on this method, we could simulate the missing OCO-2 satellite
observations in large areas and evaluate the spatial prediction ability of the stacking regression model.

### 2.3.3 Temporal extension validation

The existing studies mainly concentrated on the same period for the model training and validation, and ignored the stability of
the model in the temporal dimension. This means that it is difficult to determine the performance of the model in different
years. Therefore, temporal extension validation was designed to verify the decay performance of the stacking regression model
in the temporal dimension, to ensure consistency of the product quality. Here, the matched OCO-2 data from 2019 to 2020
were used to assess the stacking regression model trained by data from 2015 to 2018.

### 2.3.4 Ground station observation validation

The XCO$_2$ accuracy measured by the TCCON stations is constrained with a precision better than 0.25% (1-sigma) under clear
or partly cloudy skies (Messerschmidt et al., 2011), which is approximately less than 0.5 ppm (Mostafavi Pak et al., 2023), so
it is suitable to use TCCON XCO$_2$ data to quantitatively evaluate the prediction deviation of the stacking regression model. In
this study, the TCCON station observations were averaged to an 8-day resolution and matched to the corresponding 0.05-
degree grid. In order to eliminate the potential impact of the satellite observations on the accuracy of the ground station
validation, we removed the records with both station observations and OCO-2 satellite estimations. Finally, a total of 6291
records from 30 stations were obtained for the ground station observation validation.



# 3 Results and analysis


The 10-fold cross-validation, spatial expansibility validation, temporal extension validation, and ground station observation
validation were implemented to assess the performance of the stacking regression model and GCXCO$_2$ product. The results of
these validation methods are presented in turn in this section. The annual and seasonal distribution of global XCO$_2$ is then
explored, and the long time-series XCO$_2$ changes at a global scale and different latitudes are analyzed.

## 3.1 Ten-fold cross-validation result


The training dataset was divided into 10 subsets for the 10-fold cross-validation, and then each subset was validated separately.
All the validation results are summarized in a scatter plot (Fig. 4), where the overall results show a high accuracy, with the R$^2$
equal to 0.974 and the RMSE equal to 0.551 ppm. The high R$^2$ and low RMSE show that the stacking regression model has
an excellent fitting ability on the full training dataset. In addition, the R$^2$ and RMSE of each validation result are very close
(Table S1), indicating that the trained stacking regression model is very stable and there is no overfitting of the model. The
regression slope of the trend line is 0.97, which is very close to 1, further indicating good consistency between the predicted
values and the OCO-2 XCO$_2$. Therefore, the stacking regression model performs well in the 10-fold cross-validation, showing
a high ability for XCO$_2$ prediction, with only small deviation between the predicted and OCO-2 values.

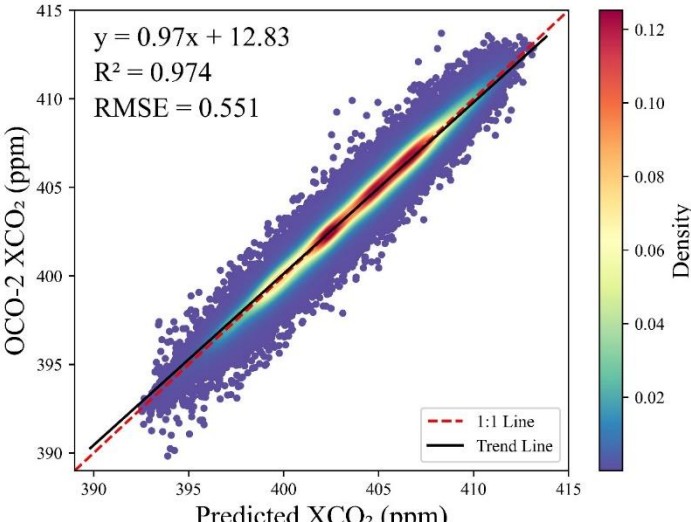


**Figure 4.** The overall results of the 10-fold cross-validation (all the validation results are summarized in a scatter plot).

## 3.2 Spatial expansibility validation result


The global region was divided into 23 areas according to the shape of the satellite observations for the spatial expansibility
validation, which allowed us to verify the prediction accuracy for areas without satellite observations. The results show an
average R$^2$ of 0.958 and an average RMSE of 0.692 ppm (Fig. 5) in the 23 regions. The maximum R$^2$ is 0.979 and the minimum



$R^2$ is 0.937, and the RMSE ranges from 0.451 to 0.877 ppm. These results all demonstrate a good accuracy with minimal
differences, which proves that the stacking regression model has a strong generalization ability in different regions. Even if
there are no satellite observations in an area, the accuracy of the prediction results is still good. At the same time, the areas
with higher $R^2$ typically have lower RMSE, and are mainly distributed in the ocean areas, ranging from 140 °E to 180 °E, 60 °W
to 40 °W, and 140 °W to 180 °W. In contrast, the regions with high RMSE have higher continental proportions, primarily ranging
from 120 °W to 80 °W, 20 °W to 20 °E, and 60 °E to 100 °E. Previous studies (Connor et al., 2016) have calculated that the total
$XCO_2$ error of the OCO-2 satellite over ocean is usually smaller than that over land, which may be the reason for the relatively
poor accuracy in regions with strong sea-land cross-heterogeneity. Taking the area from 20 °W to 20 °E as an example, the
overall validation accuracy of this area is satisfactory, but its land proportion is relatively high, resulting in a slightly lower
overall $R^2$ than the areas with a higher proportion of ocean. In summary, there is little difference between the results for the
different strips, and they all show a good accuracy, indicating that the stacking regression model shows a stable spatial
generalization ability.

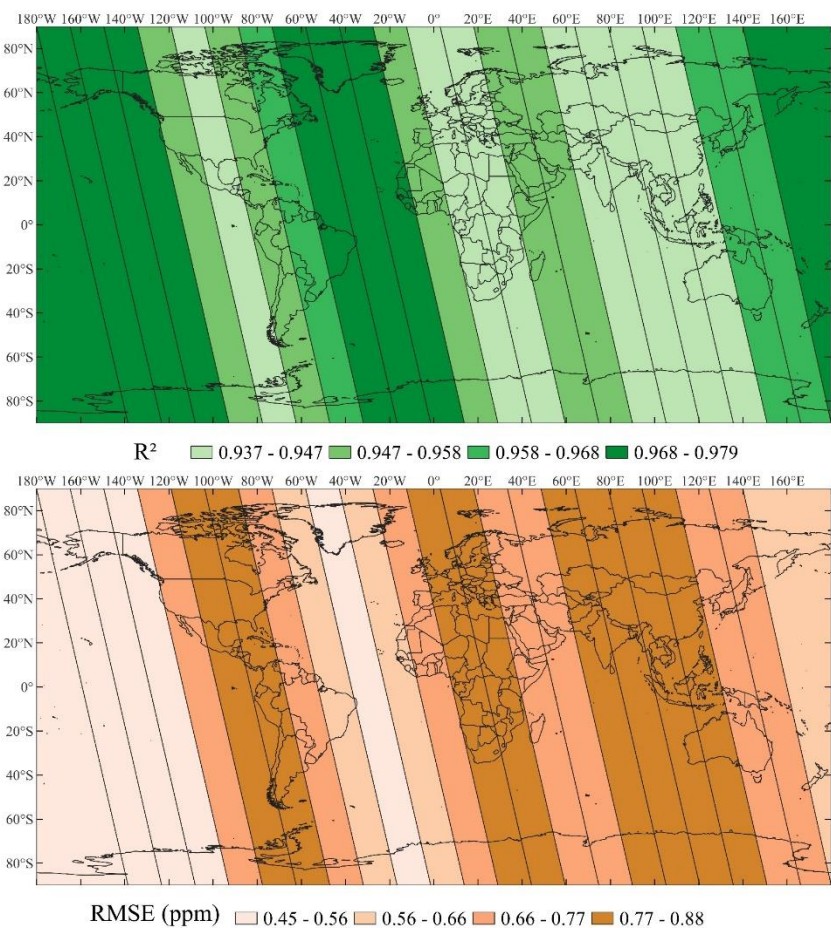






**Figure 5.** The results of the spatial expansibility validation, which represent the accuracy of each region (the solid line divisions)
being verified separately.

### 3.3 Temporal extension validation result

The matched OCO-2 data from 2019 to 2020 were used to evaluate the temporal extension performance of the trained stacking
regression model. The OCO-2 satellite has few observation samples over 8 days, so it is necessary to assess the prediction
accuracy of the trained stacking regression model during periods with few or no satellite observations. The validation results
are still good, with $R^2 = 0.886$ and RMSE = 0.823 ppm. The $XCO_2$ from the different sources between 2019 to 2020 is
compared in Fig. 6. The results (Fig. 6 (a), (b), and (c)) show that the predicted $XCO_2$ obtained using dynamic normalization
has the highest $R^2$ and the lowest RMSE, compared to the model simulation $XCO_2$. The CT $XCO_2$ has numerous discrete
points in the 395–405 ppm range, and there is a phenomenon of underestimation in the 410–420 ppm range. The accuracy of
the CAMS $XCO_2$ is generally lower than that of the CT $XCO_2$. The CAMS data are underestimated in the 400–410 ppm range
and overestimated in the 410–420 ppm range. In contrast, the predicted results are more consistent with the trend of the OCO-
2 satellite observations, with a trend line slope near 1 and RMSE less than 1 ppm. This fully proves that the predicted results
are superior to the model simulation data in the quantitative evaluation and are closer to the satellite observation level.
In order to prove the necessity of using the dynamic normalization strategy, the result obtained without adopting this strategy
is demonstrated in Fig. 6 (d). This indicates that the model without using dynamic normalization cannot predict high $XCO_2$
values correctly because the corresponding labels are not learned during the model training. Moreover, the model also cannot
deal well with discrete points ranging from 400 to 405 ppm. There have been data-driven studies (Zhang and Liu, 2023) that
have attempted to integrate multiple satellite data sources to expand the label range and avoid this phenomenon. However, it
is difficult to ensure the consistency and high accuracy of the label data, and this has not truly solved the problem of inaccurate
prediction caused by the label range. In contrast, the results obtained in this study (Fig. 6 (c)) show that the dynamic
normalization strategy can effectively solve the problem of not being able to predict values beyond the training label range. In
addition, the use of this strategy makes the model have good robustness in terms of temporal extension, and the prediction
accuracy is higher than that of the model simulation data.



Earth System
Science
Data

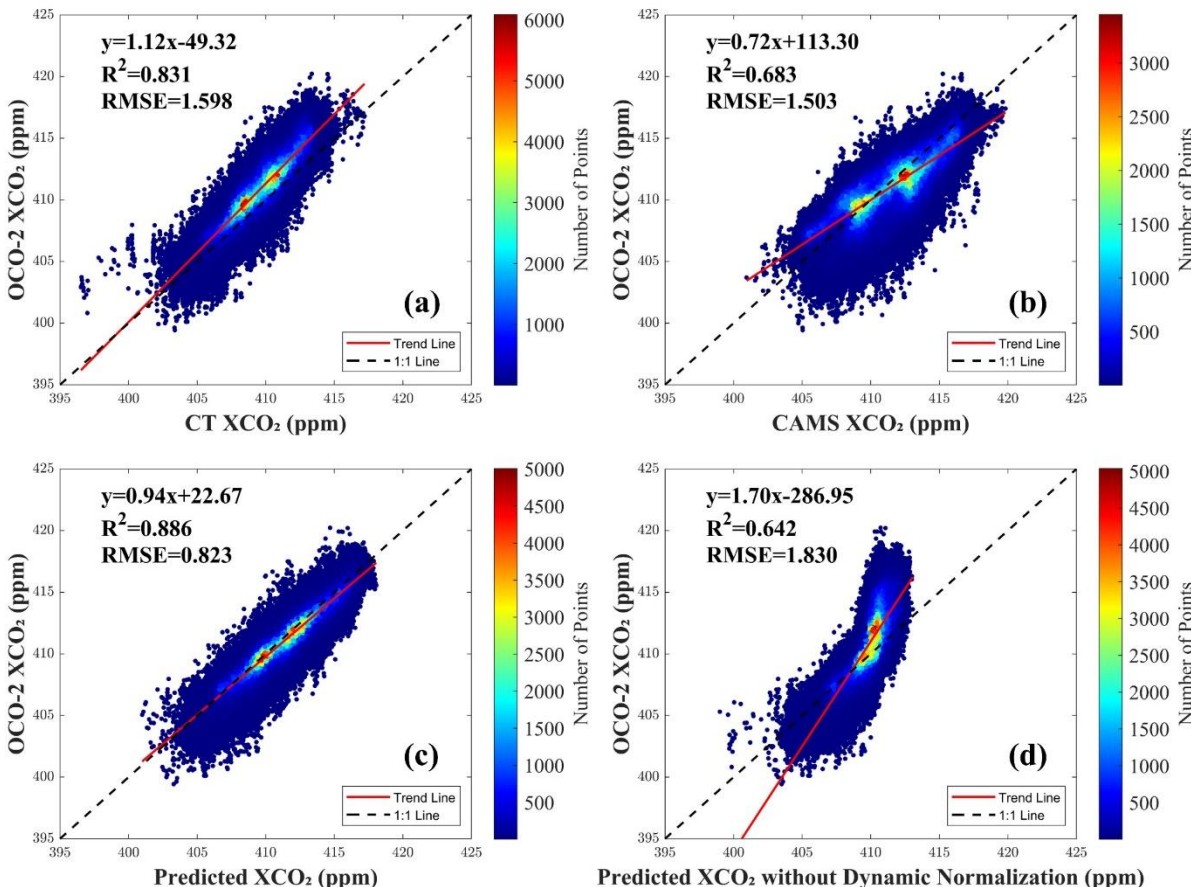

**Figure 6.** Comparison of the $XCO_2$ from different sources between 2019 to 2020: (a) CT vs. OCO-2, (b) CAMS vs. OCO-2, (c) $XCO_2$ predicted using dynamic normalization vs. OCO-2, (d) $XCO_2$ predicted without using dynamic normalization vs. OCO-2.

**3.4 Ground station observations validation result**

3.4.1 Individual station validation result

All the TCCON station observations records from 2004 to 2020 were compared with the corresponding prediction values. The results show a high-precision result with an average $R^2$ of 0.947 and an average RMSE of 1.064 ppm (Table 2). In detail, there are 27 stations with $R^2$ greater than 0.90 and 14 stations with RMSE less than 1 ppm. The PA station has the highest $R^2$ of 0.994 and the DF station has the lowest RMSE of 0.584 ppm. The $R^2$ values of the FC, MA, and XH stations are relatively low, but their RMSE values are all less than 2 ppm. This may be due to the significant changes in $XCO_2$ within a 0.05-degree grid, so that the station observations cannot represent the characteristics of this region. Meanwhile, the small number of station observations may also contribute to the low $R^2$ values. To sum up, the accuracy of the TCCON station validation is satisfactory,





with high correlation and little error between the predicted and observed $XCO_2$. The scatter plot results for each station are
included in the supplementary material (Fig. S6).
**Table 2.** TCCON station validation results from 2004 to 2020. The $R^2$ and RMSE were calculated from the station observation
records and the stacking regression model predictions.

| Station | Location | $R^2$ | RMSE | Station | Location | $R^2$ | RMSE |
|---|---|---|---|---|---|---|---|
| BR | Bremen, Germany | 0.987 | 1.659 | LH | Lauder, New Zealand | 0.947 | 1.590 |
| BU | Burgos, Philippines | 0.962 | 0.679 | LL | Lauder, New Zealand | 0.976 | 0.644 |
| CI | Caltech, USA | 0.981 | 0.99 | LR | Lauder, New Zealand | 0.909 | 0.776 |
| DF | Dryden, USA | 0.991 | **0.584** | MA | Manaus, Brazil | 0.634 | 0.831 |
| ET | East Trout Lake, Canada | 0.980 | 0.795 | NI | Nicosia, Cyprus | 0.900 | 1.120 |
| EU | Eureka, Canada | 0.992 | 1.805 | NY | Ny-Ålesund, Svalbard | 0.993 | 1.353 |
| FC | Four Corners, USA | 0.867 | 0.869 | OC | Lamont, OK (USA) | 0.989 | 0.847 |
| GM | Garmisch, Germany | 0.987 | 1.477 | OR | Orléans, France | 0.988 | 1.129 |
| HF | Hefei, China | 0.929 | 1.246 | PA | Park Falls, WI (USA) | **0.994** | 1.003 |
| IF | Indianapolis, IN, USA | 0.961 | 0.808 | PR | Paris, France | 0.962 | 1.079 |
| IZ | Izaña, Tenerife | 0.989 | 0.825 | RA | Reunion Island | 0.984 | 0.611 |
| JC | JPL, Pasadena, CA, USA | 0.937 | 0.638 | RJ | Rikubetsu, Japan | 0.958 | 1.222 |
| JF | JPL, Pasadena, CA, USA | 0.981 | 1.028 | SO | Sodankylä, Finland | 0.993 | 1.075 |
| JS | Saga, Japan | 0.980 | 0.996 | TK | Tsukuba, Japan | 0.936 | 1.335 |
| KA | Karlsruhe, Germany | 0.967 | 1.087 | XH | Xianghe, China | 0.764 | 1.819 |

3.4.2 Overall comparison between CT, CAMS, and $GCXCO_2$
The CT data, CAMS data, and the prediction results obtained in this study were compared with all the station observations.
Overall, the prediction results have the highest $R^2$ and the lowest RMSE from 2004 to 2020 (Fig. 7 (a), (b), and (c)). The
RMSE of the prediction results decreases by approximately 0.502 ppm and 0.677 ppm when compared to the CT and CAMS
data, respectively. Despite the high $R^2$ between CT, CAMS, and the station observations, there is a slight improvement in the
$R^2$ of the prediction results. From the scatter plot distribution, there is an overestimation of CAMS data from 375 to 390 ppm
in Fig. 7 (b), and a slight underestimation of CT data from 400 to 415 ppm in Fig. 7 (a). In contrast, the prediction results
alleviate the overestimation and underestimation problem of the model simulation data, with fewer discrete points. The
comparison of the model training periods (Fig. 7 (d), (e), and (f)) and model extrapolation periods (Fig. 7 (h), (i), and (j)) also
shows that our prediction results can significantly reduce the error of the model simulation data.

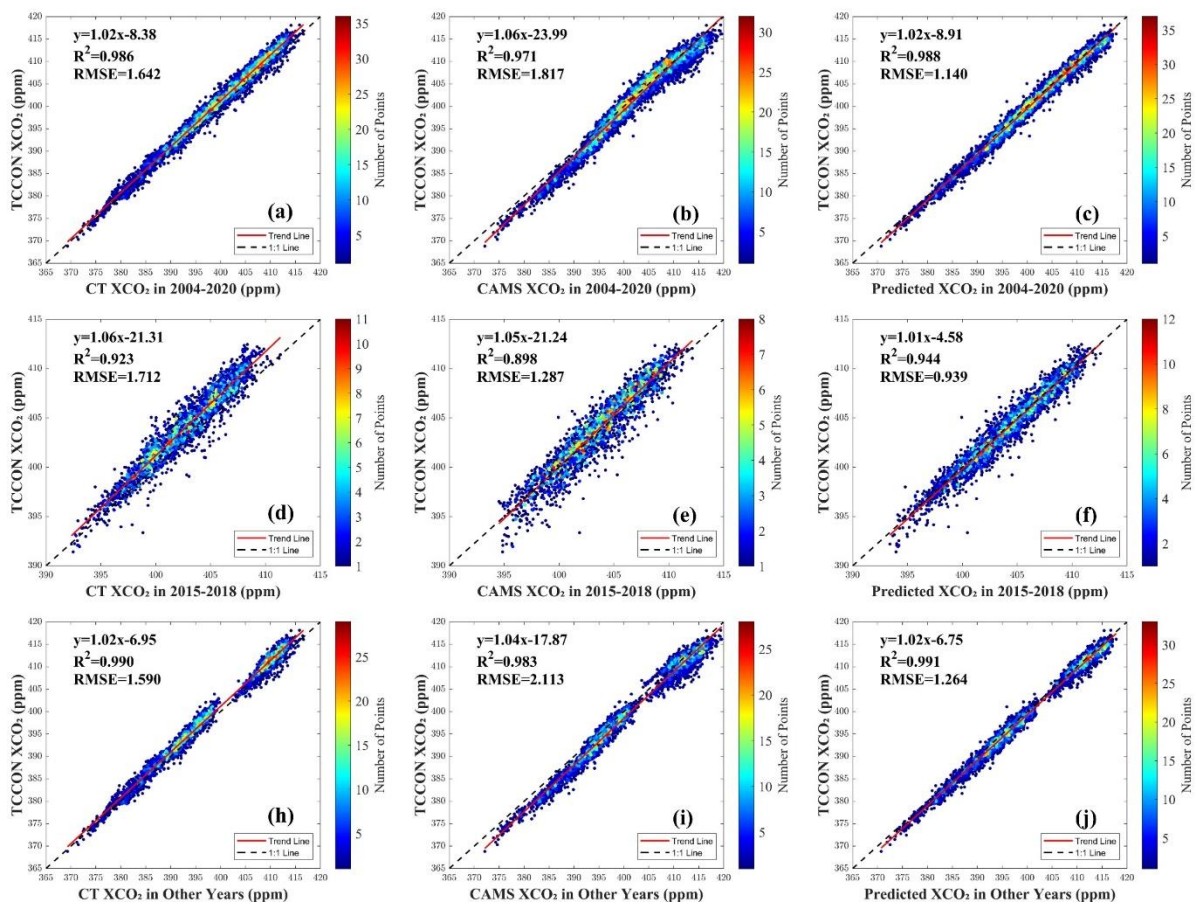

**Figure 7.** All the TCCON station observations vs. the CT data, CAMS data, and the prediction results obtained in this study, from 2004 to 2020. The data for (a), (b), and (c) are from 2004 to 2020. The data for (d), (e), and (f) are from 2015 to 2018, which is the period for model training. The data for (h), (i), and (j) are from the other years, which is the period for model extrapolation.

### 3.4.3 Yearly comparison between CT, CAMS, and GCXCO$_2$

Furthermore, the CT data, CAMS data, and the prediction results obtained in this study were compared with the station observations from different years to verify the accuracy of the product for each year (Fig. 8). Firstly, it is clear that the CAMS data have a relatively high $R^2$ from 2004 to 2012 and a relatively low $R^2$ from 2013 to 2020. However, the RMSE of the CAMS data is relatively high from 2004 to 2012 and relatively low from 2013 to 2020. This shows that the CAMS data quality varies greatly over the time series. Secondly, the $R^2$ and RMSE of the CT data vary relatively little in different years, with a stable data quality and performance that is superior to the CAMS data. Compared with the CT and CAMS data, the prediction results obtained in this study have the highest $R^2$ and the lowest RMSE in most years, which shows that the prediction results also have a significant advantage in the temporal dimension.



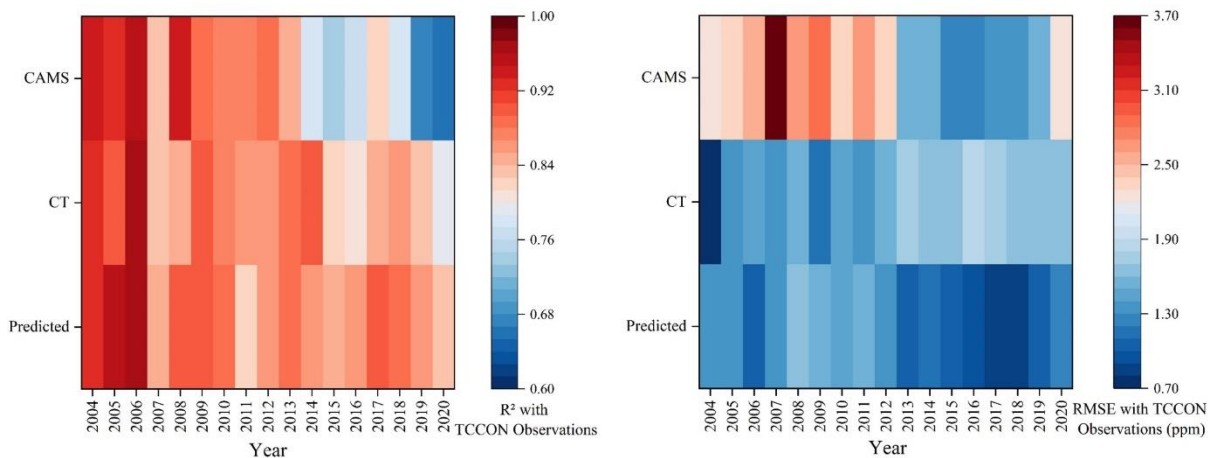

**Figure 8.** All the TCCON station observations vs. the CT data, CAMS data, and the prediction results obtained in this study, based on different years from 2004 to 2020.

## 3.5 XCO₂ spatio-temporal analysis

3.5.1 XCO$_2$ annual and seasonal distribution

We analyzed the global distribution characteristics of XCO$_2$ in 2000, 2010, and 2020 (Fig. 9 (a), (b), and (c)). The global mean XCO$_2$ for these three years is 368.63 ppm, 388.18 ppm, and 411.52 ppm, respectively. The global XCO$_2$ distribution is very similar in these years and the high-value areas of XCO$_2$ in these three years primarily distributed between the equator and 40 N. The high XCO$_2$ values on land are mainly in South-East Asia, Central Africa, southern North America, and northern South America. The low XCO$_2$ values are mainly found in the Southern Hemisphere, Outer Mongolia, and Greenland in the Northern Hemisphere. Figure 9 (d), (e), and (f) shows the trend of XCO$_2$ changes from 2000 to 2010, from 2010 to 2020, and from 2000 to 2020, respectively. Overall, the global XCO$_2$ growth rate from 2000 to 2020 was between 2.06 and 2.22 ppm. The growth rate of XCO$_2$ in the first decade (2000 to 2010) was between 1.91 and 2.13 ppm, while the growth rate in the second decade (2010 to 2020) was between 2.28 and 2.42 ppm. This indicates that the growth rate of XCO$_2$ has increased on the global scale in the past decade. At different time periods, the difference in global XCO$_2$ growth rate is not significant, and regions with slightly higher XCO$_2$ growth rates are mainly in East Asia, Central Africa, and South America.

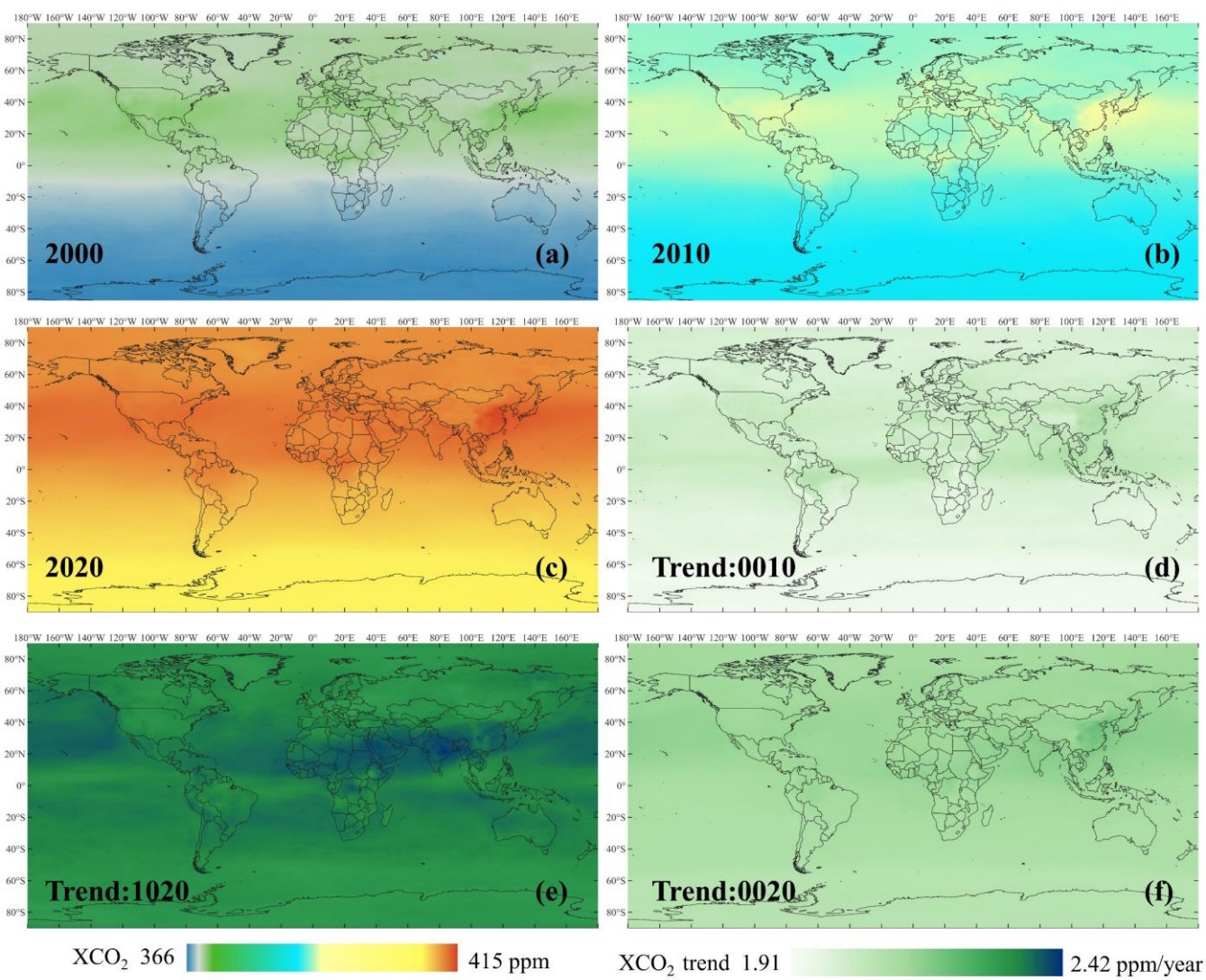

**Figure 9.** Global annual global $XCO_2$ mean distribution and trend. (a), (b), (c) represent annual global $XCO_2$ mean distribution in 2000, 2010, and 2020, respectively. (d) represents the trend of $XCO_2$ changes from 2000 to 2010, (e) represents the trend of $XCO_2$ changes from 2010 to 2020, and (f) represents the trend of $XCO_2$ changes from 2000 to 2020.

We further analyzed the distribution of $XCO_2$ in the different seasons of spring (March, April, May), summer (June, July, August), autumn (September, October, November), and winter (December, January, February). The average value of each season for 21 years is shown in Fig. 10. The high $XCO_2$ values are mainly seen in spring and winter, while the low $XCO_2$ values are mainly found in summer and autumn. The region from 40°N to 40°S is a high-value region during summer and autumn. During spring and winter, there is a significant difference in $XCO_2$ between the Northern and Southern hemispheres,



roughly divided by the equator, which may be due to two factors. Firstly, the decrease in vegetation quantity in spring and
winter leads to a decrease in $CO_2$ absorption by the ecosystem. Secondly, human activities at this time consume more energy,
leading to significant $CO_2$ emissions. In spring, $CO_2$ concentrations are higher in East Asia, South Asia, Central Africa, Central
America, and Europe. In summer, $CO_2$ concentrations in Russia, Canada, and Europe are relatively low, while concentrations
in other regions on land are similar. When it comes to autumn, Singapore, Indonesia, Brazil, the eastern United States, and
eastern China are regions with relatively high $CO_2$ concentrations. The distribution of $CO_2$ in winter is similar to that in spring,
but the concentration of $CO_2$ in the marine areas of the Northern Hemisphere is relatively low. In general, the annual
distribution of $CO_2$ is similar in spring and winter, indicating that spring and winter have a significant impact on $CO_2$
concentration during the year.

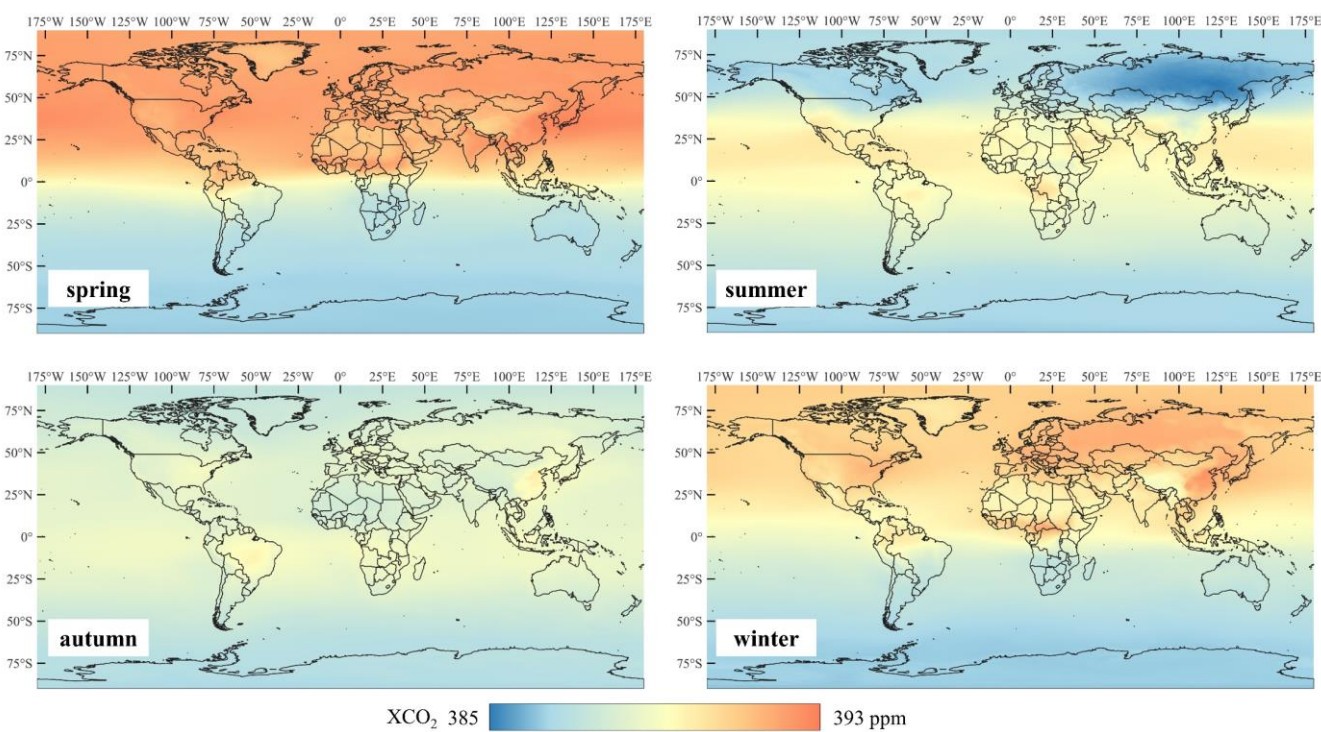


**Figure 10.** Seasonal distribution of global average $XCO_2$ from 2000 to 2020.
3.5.3 $XCO_2$ long time-series change
The global average change in $XCO_2$ every 8 days from 2000 to 2020 is shown in Fig. 11 (a). The fitting trend line indicates
that global $XCO_2$ has shown an upward trend, with an average increase of 0.0458 ppm every 8 days and an annual increase of
approximately 2.09 ppm. This reveals the significant increase in atmospheric $CO_2$ concentration from 2000 to 2020, which
may be due to human activities and the burning of fossil fuels (Jiang et al., 2022). At the same time, the global $XCO_2$ has
shown an obvious seasonal trend, showing an increasing trend from January to March and September to December, and a



downward trend from April to August. The beginning of April has seen the highest $XCO_2$ of the year, while the beginning of
September has seen the lowest $XCO_2$ of the year. The seasonal changes of $XCO_2$ each year may be related to plant growth
(Yuan et al., 2018). In winter, when plant growth slows and photosynthesis decreases, $CO_2$ concentrations in the atmosphere
usually rise slightly. In summer, the growth of plants increases and they absorb more $CO_2$, causing the concentration of $CO_2$
in the atmosphere to decrease. Based on the distribution of the concentrations around the trend line, it can be seen that the $CO_2$
growth rate from 2000 to 2008 was close to the annual average of 2.09 ppm, the growth rate from 2009 to 2015 was lower
than the annual average, and the growth rate from 2015 to 2020 was higher than the annual average.
The global region was divided into 18 regions based on a latitude bandwidth of $10°$ to analyze the temporal variation
characteristics of $XCO_2$ at different latitudes. The result (Fig. 11 (b)) indicates that the changes in $XCO_2$ at different latitudes
were similar to the global changes, and they all showed a continuous upward trend. The $XCO_2$ was close in the different
latitudes in summer, while the $XCO_2$ in the Northern Hemisphere was significantly higher than that in the Southern Hemisphere
in winter. Meanwhile, we found that $XCO_2$ value changed sharply at the equator, with significant differences between the
Northern and Southern Hemisphere in winter.

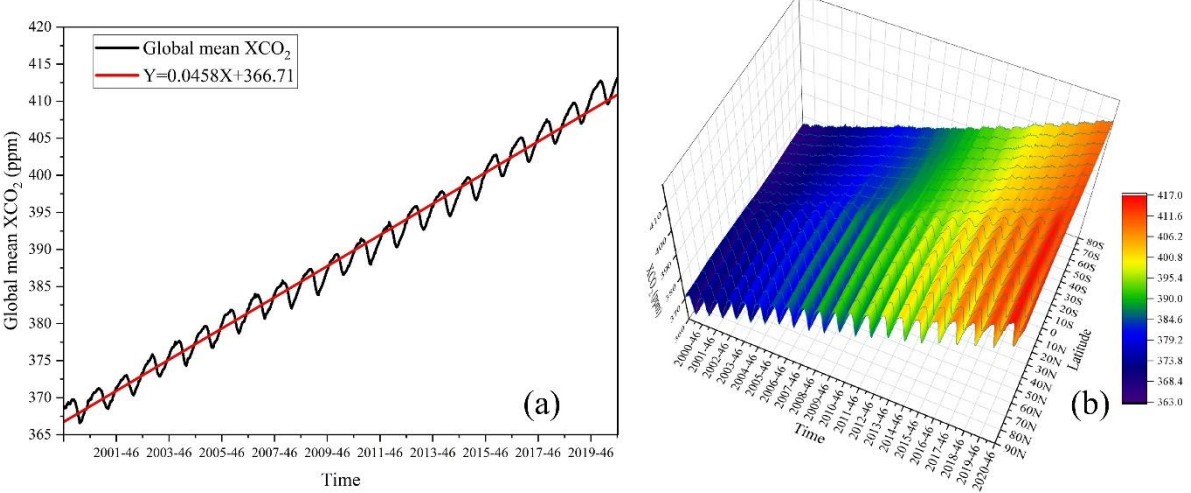

**Figure 11.** $XCO_2$ long time-series change: (a) Changes in global average $XCO_2$ every 8 days from 2000 to 2020. 2001-46
represents the 46th cycle of 2001 (eight days per cycle). (b) Long-term change of $XCO_2$ at different latitudes (2000–2020).
**4 Discussion**
**4.1 Comparison of stacking regression and basic regression**
To verify the effectiveness of the stacking regression model, we compared the 10-fold cross-validation $R^2$ and RMSE between
stacking regression and basic regression (Fig. 12). The mean $R^2$ of the stacking regression cross-validation is 0.974, which is
better than the basic regression. At the same time, the RMSE of the stacking regression model is also the lowest, at only
0.551 ppm, indicating that the stacking regression model is very stable. Among the five basic regressors, the GBR model
performs the worst, the LGBM and XGB models have an $R^2$ greater than 0.96, while the ETR and RF models perform better.
With respect to the RMSE, the GBR model shows the worst performance, followed by the LGBM and XGB models, while the
ETR and RF models achieve better results. Overall, from the quantitative results, the stacking regression model performs the
best.

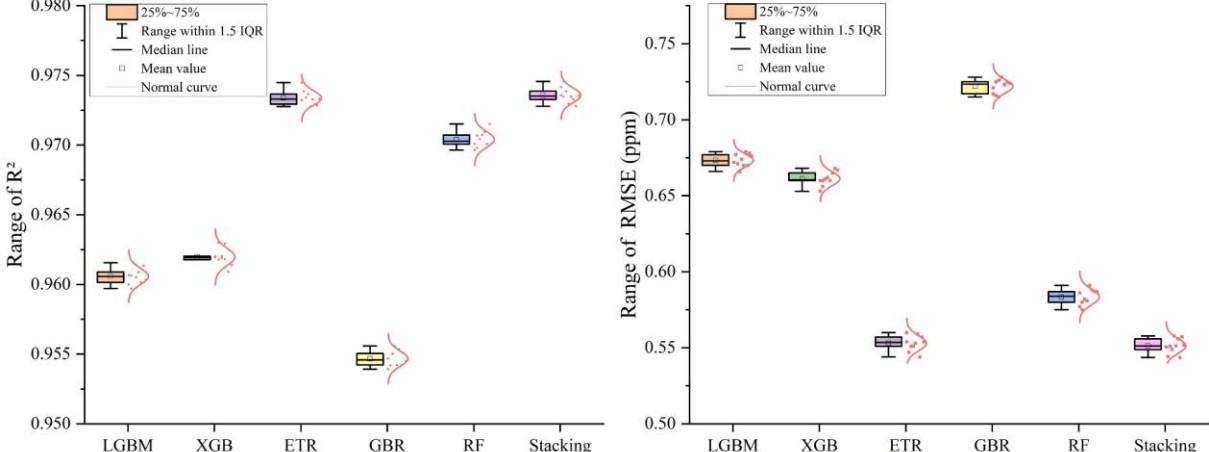


**Figure 12.** Comparison of the 10-fold cross-validation results between stacking regression and basic regression (Each point represents the result of each validation, and the curve represents the normal distribution curve; IQR refers to interquartile range of data).

Furthermore, we also compared the spatial distribution of the stacking regression product, ETR product, CAMS data, and CT
data. Typical regions from different periods were selected in Fig. 13. Overall, the stacking regression product and ETR product
have more spatial details than the CT and CAMS model simulation data, and their spatial distributions are consistent. In the
case of the 45th period of 2003, we chose East Asia for comparision and found that the spatial details of the stacking regression
product are richer than that of ETR product. Meanwhile, the spatial distributions of the stacking regression product and ETR
product are more consistent with the CAMS data in this period. In the case of the 45th period of 2020, we chose Amazon
region for comparision and found that stacking regression product still have better spatial distribution. However, the stacking
regression product in this period is more similar to CT data and differ from the situation in 2003. This may be due to the high
accuracy of CAMS data in 2003 and the low accuracy near 2019, which is consistent with our results in section 3.4.3. This
phenomenon indicates that our product fully combines the advantages of CAMS and CT data, reducing the uncertainty of
$XCO_2$ spatial distribution. Therefore, although ETR and the stacking regression model are relatively close in the quantitative
results, it is clear that the stacking regression model shows advantages in the spatial distribution, and we believe that stacking
regression is more suitable for the mapping of global high spatio-temporal resolution and high-accuracy $XCO_2$.



**Figure 13.** Comparison of the spatial distribution between the stacking regression product, ETR product, CAMS data, and CT data in different periods.

## 4.2 Variable importance analysis

In order to explore which variable has a significant impact on $XCO_2$, we used the permutation importance method to evaluate the importance of the explanatory variables. The results of this method depend on the decrease in the performance score of the



model after the variables are randomly rearranged (Breiman, 2001). The specific calculation process is as follows. Firstly,
select an evaluation index (such as $R^2$ or RMSE) for the trained model and calculate the initial score on the validation set. Then,
randomly shuffle each variable in the validation set and recalculate the corresponding score of the model. The importance of
a variable is defined as the difference between the recalculated score and the initial score.
Here, $R^2$ was selected as the evaluation index for the variable importance analysis. Each variable was randomly shuffled 10
times, and the change in $R^2$ based on the evaluation results was calculated. The results (Fig. 14) indicate that CT and CAMS
are the two main influencing variables, due to the strong correlation between the model simulation data and satellite observation
data (Mustafa et al., 2020). In this study, CT data plays a more important role than CAMS data, which may be due to the higher
correlation between the CT2022 data and the OCO-2 satellite data. In addition, we found that the SCYCLE variable causes a
0.149 change in $R^2$, indicating that $XCO_2$ has significant periodicity, which is consistent with our analysis result in Section
3.5.2. The other auxiliary variables together also can cause a 0.149 change in $R^2$, indicating that the selected auxiliary variables
can effectively supplement information for the mapping of $XCO_2$. For the meteorological data, QV has the greatest impact,
contributing to a change of 0.037. In terms of the remote sensing auxiliary data, REF6 has the greatest impact, indicating that
remote sensing data still play a certain role in $XCO_2$ mapping. Although the importance of remote sensing auxiliary data is not
very high, the spatial distribution details of the $GCXCO_2$ product are all derived from the remote sensing auxiliary data.

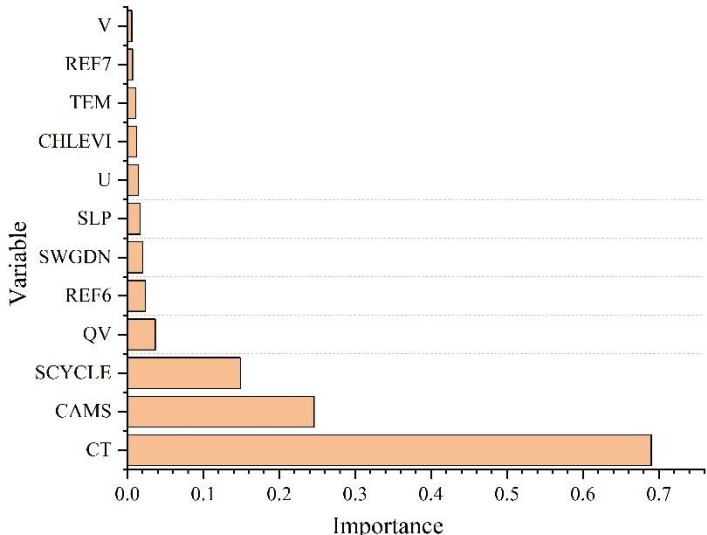


**Figure 14.** The importance of each explanatory variable calculated by the permutation importance method.





## 4.3 Innovation and limitations

In this study, a high-accuracy global $XCO_2$ dataset was generated—the $GCXCO_2$ product—with a spatial resolution of 0.05 degrees and a temporal resolution of 8 days, from 2000 to 2020. Furthermore, the newly proposed spatio-temporal validation method acted as a supplement to the existing validation methods.

The contributions of this work are as follows. Firstly, a method for global seamless $XCO_2$ mapping covering terrestrial and ocean areas was developed, based on remote sensing data, model simulation data, and meteorological data. The results demonstrate the high accuracy and stable spatio-temporal scalability of the model. Compared to the existing products (Li et al., 2022; Zhang et al., 2022; Zhang and Liu, 2023), which all cover terrestrial areas, the $GCXCO_2$ product covers both terrestrial and ocean areas and achieves a full spatial coverage. Secondly, the use of the dynamic normalization strategy in the model training effectively improves the generalization ability of the model in the temporal dimension. Due to the large range of $XCO_2$, it is almost impossible to directly use short-term concentration values to construct a model to achieve long-term inversion, and the results without using dynamic normalization show obvious errors. However, we solved the key problems by using a moving normalization method with the help of model simulations, and thus we can first achieve 21-year mapping. It means that it is possible to rely on short-term satellite observations for long-term $XCO_2$ mapping. Finally, we developed a novel validation method to evaluate the spatio-temporal extensibility in the absence of OCO-2 satellite observations. The spatial expansibility and temporal extension validations also prove the high accuracy of the $GCXCO_2$ product.

However, there are still some limitations to this work. Firstly, the global $XCO_2$ mapping method is heavily reliant on the $XCO_2$ model simulation data, which limits the real-time production ability. Future research should attempt to utilize more suitable remote sensing explanatory variables for real-time mapping. Meanwhile, we only used OCO-2 satellite observations in this study, and future studies could use multiple satellite data sources to obtain more samples, which would involve multi-sensor fusion and put forward a higher requirement for data processing. Finally, although the various validation methods have confirmed the high accuracy of the stacking regression model and product, we were unable to analyze the authenticity of the spatial distribution of the product, due to the lack of real high-resolution seamless $XCO_2$ data. Therefore, exploring validation methods for the spatial distribution is also a potential research direction.

## 5 Data availability

The long-term (2000-2020) global $XCO_2$ dataset $GCXCO_2$ can be obtained freely at https://doi.org/10.5281/zenodo.10083102 (Guan and Sun, 2023). The data is stored in NetCDF file format, with a time resolution of 8 day and a spatial resolution of 0.05 degree. The file is named after "year-cycle", for example, 2000-01 represents the $XCO_2$ data for the first eight day of 2000.



## 6 Conclusion

In this study, the stacking regression model was utilized to construct the nonlinear relationships between the OCO-2 satellite $XCO_2$ data and satellite observations, model simulation data, and meteorological data for global seamless $XCO_2$ mapping. The high spatio-temporal resolution (8-day, 0.05 degree) global $GCXCO_2$ product covering 2000 to 2020 was produced. The 10-fold cross-validation results ($R^2 = 0.974$, RMSE = 0.551 ppm) and the TCCON station validation results ($R^2 = 0.988$, RMSE = 1.140 ppm) confirmed that the model and product have an overall good performance and accuracy. Furthermore, the results of the spatial expansibility validation ($R^2 = 0.958$, RMSE = 0.692 ppm) and temporal extension validation ($R^2 = 0.886$, RMSE = 0.823 ppm) also demonstrated that the stacking regression model has an excellent spatio-temporal generalization ability. The innovative use of dynamic normalization enabled the model to expand in the temporal dimension and successfully generated the product covering 21 years. More importantly, the comparison at different scales proved that the $GCXCO_2$ product has a higher spatial resolution and accuracy than the model simulation data, and is closer to the accuracy level of the OCO-2 satellite data.

According to the $GCXCO_2$ product, the seasonal distribution of global $XCO_2$ varies significantly, and the $XCO_2$ in the Northern Hemisphere is clearly higher than that in the Southern Hemisphere in spring and winter. Meanwhile, from 2000 to 2020, the global mean $XCO_2$ has risen from 368.65 ppm to 411.49 ppm, indicating an average annual increase of approximately 2.09 ppm and revealing apparent global-scale changes. The $XCO_2$ at different latitudes has also shown a similar upward trend and seasonal variation characteristics in the long time series.

The $GCXCO_2$ product generated in this study will be of great significance for regional carbon monitoring, carbon policy formulation, and global carbon flux calculations, and can also provide seamless $CO_2$ data for global climate change studies, ecological research, and other studies.

**Author contribution**

**Huanfeng Shen:** Conceptualization, Methodology **Xiaobin Guan:** Project administration ,Writing - Review & Editing, Data curation **Zhihao Sun:** Writing - Original Draft, Validation, Visualization, Software **Dong Chu:** Methodology **Guanglei Xie:** Validation, Software **Yuchen Wang:** Software, Resources

**Competing interests**

The authors declare that they have no conflict of interest.

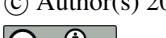


## Acknowledgements

The authors would like to thank USA Goddard Earth Sciences Data and Information Services Center for providing the OCO-
2 satellite products and MERRA-2 meteorological products, Global Monitoring Laboratory (GML) of the National Oceanic
and Atmospheric Administration (NOAA) for providing CT2022 $XCO_2$ products, Copernicus Climate Data Store for providing
the CAMS-EGG4 $XCO_2$ products, and the Total Carbon Column Observing Network for providing ground $XCO_2$ observation
data.

## Financial support

Our work is supported by the National Natural Science Foundation of China (42371364, 42001371), and the Open Fund of
Hubei Luojia Laboratory (220100041).

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
