# Peer review of "A long-term (2000-2020) global 0.05 ° continuous atmospheric carbon dioxide dataset (GCXCO2) combining OCO-2 observations and model simulations based on stack learning"

_Earth System Science Data, 2023_

## Referee Comment (RC1)

Guan et al. utilize an ensemble machine learning method to estimate a long-term global atmospheric column carbon dioxide dataset based on multi-source data. The comprehensive validation of predicted $XCO_2$ confirmed the generalization of the model and the reliability of the dataset. In particular, the dynamic normalization strategy significantly improves the performance of the model. The application of the dataset reveals significant seasonal distribution and long-term changing trends in global XCO2. However, there are some issues that the authors need to address before the manuscript can be considered for publication. The specific comments are listed below.

1. L77-78: The data on ocean area is the highlight of the study in the Introduction, but in the text, in addition to considering the characteristics of CHL-a as a variable, there is no verification and application analysis for the ocean. OCO-2, carbon tracker, and CAMS are also covered in marine areas. It is suggested that the authors reorganize this part to better highlight the innovation aspect.

2. L175, 177: What does "each period" refer to? It is for each year?

3. L184: In Step 3, were the variables of CAMS and CHLEVI removed to train another model based on the training dataset between 2015 to 2018 for the mapping from 2000 to 2002?

4. L312: why the validation results of the supplementary model (SR0002), which removed the CAMS and CHLEVI variables, perform better ($R^2$ is 0.898 in Fig. S3c, and slope is 1.27 in Fig. S3d) in the temporal extension validation compared to the main model ($R^2$ is 0.886 in Figure 6c, and slope is 1.79 in Figure 6d). It seems that model SR0002 can better solve the problems of low-valued and high-value overvaluation of machine learning models. It is also confirmed by the comparison between Figure 7c, j and Fig. S4c, j (with lower RMSE values). Perhaps model SR0002 has a better performance for the prediction with temporal extension. I suggest that the ranges of the color bar between Figure 5 and Fig. S2 be consistent for the comprehensive comparison.

5. Fig. S5, 6: What is the time range in the validation for each TCCON station of models SR0002 and SR0320?

6. L237: Why are there only 150000 data that were randomly selected from other regions instead of all data to train the model for the spatial expansibility validation?

7. L276-L279: It is suggested that the spatial expansibility verification can provide evaluation results on a grid scale, rather than on the shape of the satellite observations for supporting the results of the continental regions with high RMSE and lower $R^2$.

8. L174: The dynamic normalization strategy has a surprising improvement for the modeling. Is this the author's original contribution? If so, please provide the relevant background in the introduction. If not, please add a reference. And why is the performance of predicted $XCO_2$ without dynamic normalization worse than that of the input variables (CT $XCO_2$ and CAMS $XCO_2$) between 2019 to 2020 (Figure 6)? What is the result of the training set (2015-2018)? Are the hyperparameters of the model with dynamic normalization the best? Is there any overfitting in this model?

9. Section 3.4.3 and L252, I'm curious about the reason for the higher $R^2$ in the extrapolation periods compared to the training periods. The annual verification accuracy between TCCON station observations and OCO-2 satellite estimates in matching grids may explain this.

10. L369: Please add the calculation method of trends, and the statistically significant (P values).

11. Section 4.1: The evaluation indicators and spatial distribution between Stacking and ETR models are very similar. Can authors provide additional evidence that the stacking model yields better prediction, such as the ability to predict extreme values, the spatial difference between the stacking model and the ETR model, and so on?

12. L468: I suggest that authors add a comparison table between the predicted dataset and previous studies in the Supplement, summarizing spatiotemporal resolution, time span, whether it contains the ocean, verification method and accuracy, etc. Especially, the comparison of the ocean and resolution can support your Introduction.

13. L475-476: The expression "we developed a novel validation method to evaluate the spatiotemporal extensibility …" may not be accurate enough. As far as I know, many atmospheric studies have used similar temporal and spatial evaluation methods.

---

## Referee Comment (RC2)

In this study, the authors aimed to create a method for fixing gaps in satellite carbon dioxide data. Although this is valuable, I believe it falls short of article publication. They filled in missing CO2 data using extra satellite info and simulations. Here are the main problems:

1. The study treats satellite CO2 as observations. While satellite data can replace actual observations in some cases, it must fully match observations. Past studies found significant differences between oco-2 measurements and ground observations.

2. Before building the model, CT assimilation system simulations closely matched observations. The model's high accuracy is because CT's CO2 simulation was precise. So, developing a machine-learning model just to reduce a not-so-significant error rate doesn't make sense.

3. To show the model's quality, the authors used ground-based data. CAMS and CT assimilation systems use ground data to improve predictions. CT simulations matching observations well is likely because these systems previously used ground-based data to correct simulations.

4. Similarly, satellite CO2 is used in assimilation systems for correction. The CT and CAMS simulations used as inputs depend on the machine learning model's output (satellite CO2). This dependency raises questions about the model's reliability.

5. Creating a machine learning model for satellite data using dynamic model simulations contradicts the main advantage of statistical models—low computational cost. In reality, implementing this model means getting outputs from both CT and CAMS, undermining its intended efficiency.

While there are additional aspects to consider, fundamental issues in developing the model for this study hinder further exploration. Regrettably, given these challenges, I anticipate difficulties in publishing this article in a reputable journal such as ESSD.

---

## Referee Comment (RC3)

Not suited for ESSD.

Well written. Good data access at Zenodo. Authors apply multiple ML techniques. But effort as described fails to meet internal metrics and journal expectations. Recommend rejection.

Authors propose to share "the first remote sensing-based global high-precision long-term XCO2 dataset". Not true by a long shot. A large contingent of researchers, inside and out of NASA, seek to explore and confirm OCO-2 measurements of XCO2. This manuscript unfortunately reports very little of that other work (only Conner et al. 2016?) evaluating, among other complexities, aerosols, clouds, sensor sensitivity and degradation, orbit degradation, etc. A full estimation of accuracy (low on some scales) and precision (highly variable) for XCO2 from OCO-2 exists, not cited here. Another manuscript in ESSD (2023-449) attempts a similar evaluation; also not cited here. Latest version of TCCON also in ESSD, likewise not cited here.

Authors promise new distinction, terrestrial vs ocean. They relate downloading MODIS EVI, CHL-a, and reflectance data (lines 128 to 131). But reader finds only weak reports around lines 275 to 285, with nothing firm nor quantitative. Certainly no uncertainties, signal to noise, etc. Authors fail to meet their own expectations?

Validation seems uncertain and, unfortunately, inconclusive. Their product as good as, or closely correlated to, CarbonTracker (CT, as but one example)? But a user community knows and trusts e.g. CT (in part because of extensive reports on CT uncertainties - see below), so with extra effort (and computing) this product adds what to CT? Nothing, evidently. Authors here correlate (validate?) with both CT and other model syntheses, and with TCCON, but CT and those other simulations also validate against TCCON. So how can these authors validate against a product that also serves as reference to other products they also validate against? NOAA does not, so far as this reader knows, report any XCO2, only surface and 'above marine boundary layer' measurements. With extreme attention to precision and representativeness, both completely missing here.

What, if anything, did we gain here? Authors have not made a case for real improvements. They like to claim "spatial resolution of 0.05 degrees and a temporal resolution of 8 days, from 2000 to 2020"? Beg pardon but don't we already know that? 411 ppm for 2020? This reader can learn that easily from daily co2. Trend of 2 to 2.2 ppm per year over those decades? Again, this reader can learn that from NOAA's flask network. Plus we know that trend of atmos CO2 has changed (increased) over recent decades. But not here? If authors have made real advancement, they need to prove it. Too often they err by citing 'surface' CO2 data when in fact they report XCO2 data.

Biggest failure: no uncertainties. Authors claim "high-precision" but in fact ignore precision/ uncertainty entirely. 'Uncertainty' as a term never appears in their text. Never an error bar. Reader encounters RMSE but those assume linear correlations and apply, on their own scales, only within specific figures. OCO-2 data products come with well-described much-discussed uncertainty matrix (unfortunately ignored here). On top of that, with abundant un-cited evidence, authors insert additional uncertainty with every step and modification: cross-fold validation (introduces uncertainties); spatial expandability (introduces substantial additional uncertainties); temporal extensions (further substantial uncertainties). Who might know these uncertainties better than these authors? Reader finds and learns nothing. Personally, this reader doubts, after cumulating collective uncertainties across multiple steps, that they can claim better than $\pm$ 50% in annual XCO2 increments or $\pm$100% in 2000 to 2020 trends. Have they done better than my estimates? RMSE's do not answer these questions; author provide no evidence one way or another.

Not clear from evidence presented here that this product adds any value.